# Composting Processes for Agricultural Waste Management: A Comprehensive Review

Muhammad Waqas [1,†], Sarfraz Hashim [2,†], Usa Wannasingha Humphries [3,*], Shakeel Ahmad [4], Rabeea Noor [5], Muhammad Shoaib [5], Adila Naseem [6], Phyo Thandar Hlaing [1] and Hnin Aye Lin [1]

1   The Joint Graduate School of Energy and Environment (JGSEE), King Mongkut's University of Technology Thonburi, Bangkok 10140, Thailand
2   Department of Agricultural Engineering, MNS University of Agriculture, Multan 66000, Pakistan
3   Department of Mathematics, Faculty of Science, King Mongkut's University of Technology Thonburi, Bangkok 10140, Thailand
4   College of Environmental Science and Engineering, Nankai University, Tianjin 300350, China
5   Department of Agricultural Engineering, Bahauddin Zakariya University, Multan 60000, Pakistan
6   Department of Food Science and Technology, Bahauddin Zakariya University, Multan 60000, Pakistan
*   Correspondence: usa.wan@kmutt.ac.th
†   These authors contributed equally to this work.

**Abstract:** Composting is the most adaptable and fruitful method for managing biodegradable solid wastes; it is a crucial agricultural practice that contributes to recycling farm and agricultural wastes. Composting is profitable for various plant, animal, and synthetic wastes, from residential bins to large corporations. Composting and agricultural waste management (AWM) practices flourish in developing countries, especially Pakistan. Composting has advantages over other AWM practices, such as landfilling agricultural waste, which increases the potential for pollution of groundwater by leachate, while composting reduces water contamination. Furthermore, waste is burned, open-dumped on land surfaces, and disposed of into bodies of water, leading to environmental and global warming concerns. Among AWM practices, composting is an environment-friendly and cost-effective practice for agricultural waste disposal. This review investigates improved AWM via various conventional and emerging composting processes and stages: composting, underlying mechanisms, and factors that influence composting of discrete crop residue, municipal solid waste (MSW), and biomedical waste (BMW). Additionally, this review describes and compares conventional and emerging composting. In the conclusion, current trends and future composting possibilities are summarized and reviewed. Recent developments in composting for AWM are highlighted in this critical review; various recommendations are developed to aid its technological growth, recognize its advantages, and increase research interest in composting processes.

**Keywords:** composting; biodegradability; decomposing; organic waste; agricultural waste management

## 1. Introduction

Waste production is proportional to the number of human inhabitants worldwide. Thus, the increasing global population and continually growing human demands have resulted in massive waste production. With a population of 212 million in 2019 [1], Pakistan generates more than 20 million tons of waste annually [2]. On average in Pakistan, waste generation per capita is 0.612 kg/day; from this amount, 60 to 65% of waste is organic and biodegradable [3]. Organic matter (OM) in Pakistani soil is <1% [4]. Agricultural waste (AW) are leftovers of agricultural activity on agricultural land. Owing to lack of access to disposal sites in Pakistan, agricultural waste is frequently mismanaged; thus, most AW is burned or destroyed [3]. To protect the environment and ensure sustainable agriculture, resilient rural regions, and productive farming, it is vital to pursue the appropriate use and development of AW management (AWM). Among the different methods for

managing organic waste, such as landfilling and incineration, biological decomposition of AW is considered the most effective. Composting is a low-cost method of biological decomposition. Micro-organisms control the composting process. This process influences the physical–chemical parameters of heat, aeration, water content, C:N ratio, and pH [5,6]. Composting is an alternative AWM approach and the resulting compost can be recycled into valuable products. This method is considered the most effective—it is environmentally friendly and agronomically sound since the resulting compost can be utilized as a natural, organic fertilizer and soil nutrient source [7]. Composting has been defined by Ayilara et al. (2020) as a form of recycling in which organic waste is digested by microbial activity under regulated conditions to create valuable, ecofriendly, and environmentally friendly goods [8]. The microbial population, which includes bacteria, fungi, and worms, can also stabilize degradable OM in the compost. In addition, the features of the microbial population rely on the substrate and physical conditions, which include the substrate's wetness, temperature, and aeration. Composting is only appropriate for agricultural waste, so the performance of this procedure also depends on the properties of the waste [9]. Composting has numerous benefits, including lowering the waste volume, weight, and water content, and producing dormancy in harmful organisms [10]. The compost can therefore contribute to the enhancement of soil nutrient levels, which is required for plant growth and significantly minimizes the need for synthetic fertilizers [11]. As a result of its ability to boost the soil's organic carbon content, compost application can revitalize soils in dire need of revitalization. In addition, as a soil amendment, compost improves soil structure, water retention capacity, and tilth [12]. Composting is initiated and managed under regulated environmental conditions instead of a natural and uncontrolled process. Composting is distinguished from decomposition by its controlled process [13]. Composting requires a longer preparation time, emits a foul stench, requires a long time to mineralize, and may contain diseases that can tolerate high temperatures to some extent, i.e., thermotolerant pathogens, and contains insufficient nutrients. All of these factors have deterred farmers from employing composting as a method of sustainable agriculture. Following this, there has been abundant evidence for the invention of composting processes to manage AW.

Numerous studies have investigated various composting processes, including vermi-composting (VC), aerobic composting (AC), and anaerobic composting (AnC), to convert farm waste into farm manure [14,15]. AC is the breakdown of OM by oxygen-dependent bacteria. Composting bacteria occur naturally and thrive on the moisture that surrounds OM. Airborne oxygen diffuses into the moisture and is absorbed by the bacteria [16]. Mehta and Sirari (2018) stated that AC is the most efficient decomposition type, producing compost that matures quickly. The biological breakdown and stability of OM under conditions favorable to the multiplication and activity of thermophilic microbes results in a solid, pathogen-free product ideal for forestry and agriculture [15]. AnC is a "no oxygen" technique in which biodegradable materials are stacked in an enclosed environment. Typically, digesters are used. Anaerobic micro-organisms dominate the AnC process. These microbes produce intermediate chemicals, including hydrogen sulfide, methane, and acids, while leaving pathogens and weed seeds untouched [14]. VC is the process of using earthworms to compost biodegradable organic materials. By substantially eating all types of OM, earthworms can degrade the OM. Earthworms can consume their body weight daily e.g., earthworms weighing 0.1 kg can consume 0.1 kg of waste daily [16]. According to Barthod et al. (2018), it is a globally adopted, low-cost biological treatment procedure for the generation of biofertilizers for agricultural uses. Worms and micro-organisms are the primary agents in composting for recycling nutrients, controlling soil processes, and preserving soil fertility [10]. Recently, many investigators employed these composting processes for AWM, including Karak et al. (2013) who investigated composting rice straws, wheat straws, potato plants, and mustard stovers with fishpond substrate [7]. This process was carried out for 56 days utilizing a heap as a compost box. For all compost preparations, the compost temperature on the first day varied from 24 to 26.8 °C and climbed to 81 °C before persistence. Initial pH values ranged from 6.76 to 7.68, while total N concentration

was between 14.56 and 21.57 g/kg; the content of heavy metals was below the Indian Agriculture Ministry and Cooperation's limit. After composting, the C:N ratio ranged from 11–18 [17]. Qasim et al. (2018) improved the carbon-to-nitrogen (C/N) ratio to achieve a high composting and aeration rate and to create favorable circumstances for the process [13]. Azim et al. (2018) conducted a literature assessment of the most critical startup, monitoring, and maturity criteria for various composting techniques and input materials [12]. Farmers in developing countries must be aware of the process' aspects, effectiveness, and efficiency of composting. It is challenging to decide which composting technique is effective regarding all of elements used to maintain soil health.

Composting of AW is strongly encouraged in Pakistan as a massive amount of rubbish fills our overflowing landfills. Numerous researchers have investigated organic waste's physical and chemical features during the different composting processes. Consequently, this review article examines composting as an alternate AWM strategy. Therefore, the key objectives of the current review are to (a) highlight the best prominent features of the composting method via its phases and prominence in various wastes; (b) assist farmers, researcher, and scientists in the selection of treatments for different crops substrates and help them select a composting technique by providing a comparison between different techniques; (c) provide the comparison of composting techniques on the basis of nutrients; and (d) compare two-stage composting (AnC followed by AC) with AC, AnC, or VC alone.

## 2. Composting

Composting is the biological conversion of the solid waste of plant and animal organic materials into a fertile matrix through numerous micro-organisms, including actinomycetes, bacteria, and fungi, in the presence of oxygen. The addition of diverse microorganism in a solid waste can convert it into compost or many by-products, .g., heat, water, and $CO_2$ [17,18]. Humus is the solid and stable matrix after the microbiological process that can be usefully applied to land as an organic fertilizer to increase the fertility and structure of the soil. In ancient history, i.e., pre-Columbian Indians of Amazonia or ancient Egyptians and numerous prehistoric cultures used composting as a primitive technique for the betterment of soil. In the previous four decades, the composting technique has flourished, and its beneficial impact is illustrated with scientific research. The vulnerability and interconnection of various competing factors regarding the knowledge and process engineering of a composting matrix have been established [19–21].

Composting innovative processes were developed and employed by large- or medium-scale farmers, but they are expensive for small-scale farmers because the techniques require high-tech equipment for composting. Despite discrete processes/techniques, the crucial key points of the composting processes were indistinguishable each time, like natural, chemical, and physical characteristics. Appropriateness of distinctive input supplies and alterations and their fitting structure, substrate degradability, dampness management, energy, porosity, air space, energy adjustment, deterioration, and stabilization are needed to study and distinguish compost and composting processes [19,22].

### 2.1. Composting Stages

Composting processes undergo four stages: mesophilic, thermophilic, cooling, and finally ending with compost maturation; these stages can happen concomitantly rather than consequently [23] (Figure 2). Each stage duration depends on the mixture's inceptive framework, water content, air circulation, and microbiological composition [24,25]. During the mesophilic phase, a combination of bacteria, fungi, and actinomycetes induce the rapid metabolism of C-abundant substrates. Moreover, this is accomplished by selecting tolerable temperatures, generally within 15–40 °C, because aerobic metabolism will produce heat. Transforming the matter and air circulation decreases the temperature, for the time being, reducing the rapid decay of other organic matter. Thus, the temperature rises once again, as shown in Figure 1. In the thermophilic phase (2nd stage), temperature increases to around 40 °C, favoring mostly thermophilic bacteria, e.g., bacillus. When C compounds

are produced after substrate reduction, a modest temperature fall occurs followed by the cooling phase.

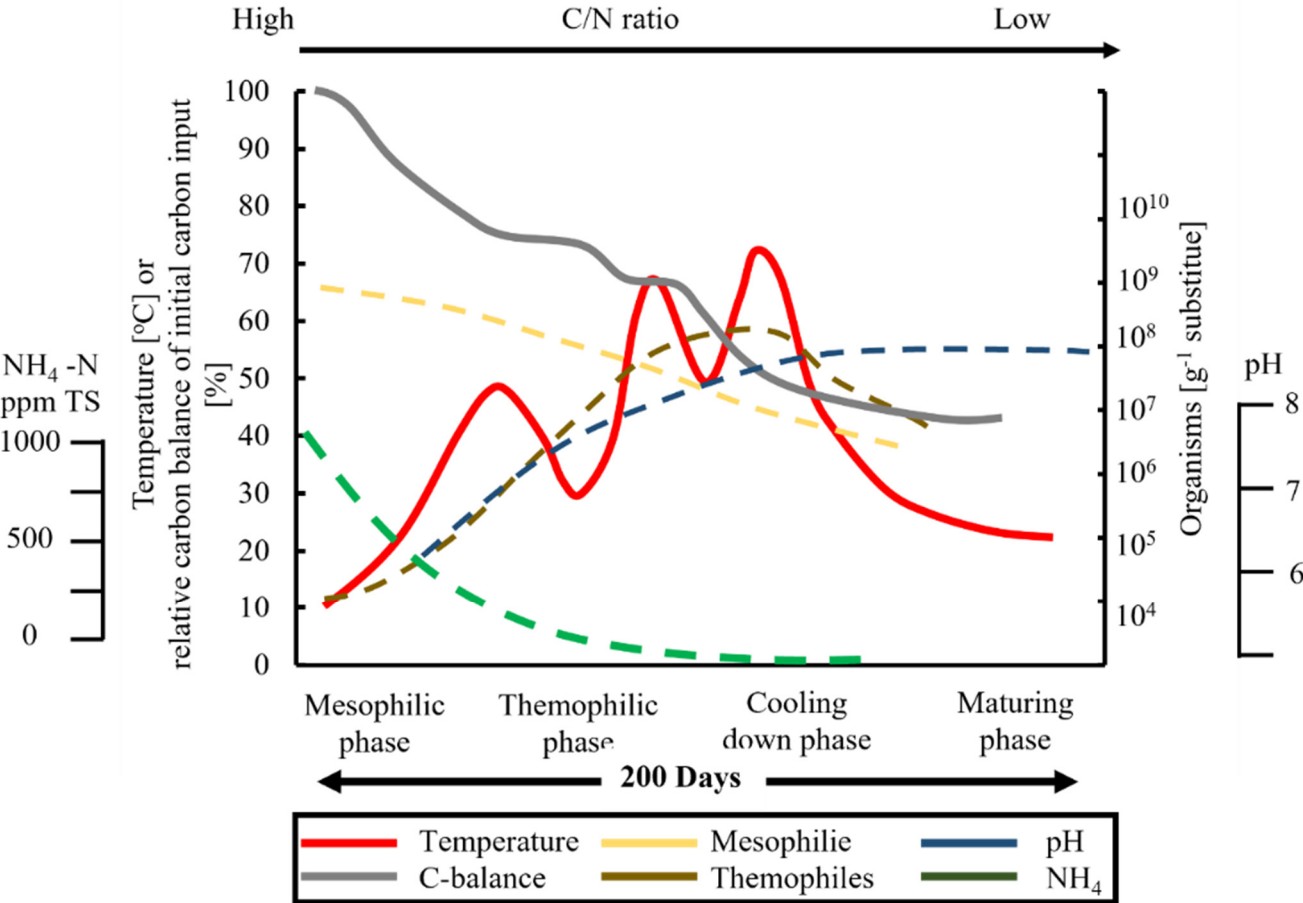

**Figure 1.** Time, temperature, the progression of compost biota, and further processes during discrete stages in composting.

Fungi break down more complex structures and more resistant components like lignin and cellulose molecules. Additionally, actinomycetes play a crucial role in forming humic compounds via condensation processes and breakdown [25]. Using aerobic bacteria, the final composting maturity is characterized by lower oxygen uptake rates and temperatures < 25 °C. During this final phase, the breakdown of various organic components continues, and macrofauna and soil organisms enter. By metabolizing phytotoxic chemicals, the organisms of this phase have a favorable effect on compost maturation, e.g., plant disease suppression [26].

Consequently, compost quality improves primarily during maturation (final stage) [27]. The final product of composting is characterized by pH and a lower C/N ratio of 15 to 20 compared to the initial substrate composition. It may contain a significant amount of plant-available $NO_3^-$, but $NO_4^+$ levels are low. Moreover, the intensity of the compost odor is significantly diminished [28]. However, it appears that the OM has stabilized, retaining recalcitrant C compounds [25]. Table 1 explains the favorable and sustainable application of different crop residues' influence on numerous biological, chemical, and physical aspects during the different processes. The outcomes showed which method is best with respect to input residues and the desired output products.

### 2.2. Discrete Waste Composting

In contrast to landfilling, which elevates the pollution risk for groundwater, discrete waste composting techniques are environment friendly and avoid groundwater contamination since chemical pollutants and bacteria are reduced during composting. Composting

permits persistent organic pollutants and endocrine disruptors to remain in the soil while beneficial bacteria break down the toxins. The elimination of these harmful chemicals has not been simple. Although numerous methods have been attempted to eradicate them, there is no agreed success rate. A thorough application can increase agricultural and environmental sustainability. It also improves soil OM content and enhances agricultural productivity [29] due to the availability of plant-growth-promoting organisms and sufficient nutrients in the composted debris [30] and significantly contributes to the certification of food safety. Compost is helpful for bioremediation [6], weed control [31], plant disease control [32], pollution anticipation [33], and erosion management, in addition to its use as fertilizer. Composting also increases soil biodiversity and reduces environmental risks associated with synthetic fertilizers [34].

Composting is a fundamental aspect of a comprehensive AWM strategy. The key strategy for practical integrated AWM is nutrition level improvement. Compost is rich in essential plant nutrients, e.g., nitrogen (N), phosphorus (P), potassium (K), sulfur (S), carbon (C), and magnesium (Mg), as well as various essential trace elements [26,35]. Consequently, compost can be described as an assortment of nutrient-rich organic fertilizers [36]. Compost processing parameters and organic feedstocks determine its key chemical features, e.g., C/N ratio and pH, as well as the content of other nutrients (Table 2). Total N, P, and K levels could contribute to soil fertility when used as soil amendment agents. By adequately combining these organic components, nutrient-rich compost substrates can be produced and used in agriculture in place of commercial mineral fertilizers. This aspect is discussed in the following subsections.

- Crop residue waste

Global agricultural waste production is substantial, and crop leftover management is imperative [37]. In addition, waste disposal pollution necessitates research into eco-friendly methods for managing agricultural wastes as the increase in agricultural waste exacerbates aesthetic, health, and environmental issues. Consequently, research into secure disposal methods is necessary. Composting has evolved into an eco-friendly, cost-effective, and secure treatment technology; it is a productive method for intensifying and preserving agricultural products [38]. Biodegradable wastes, e.g., wood shavings, pine needles, dry leaves, sawdust, and coir pith, are commingled to maintain appropriate and durable humus [39]. However, lignin-rich plant products are difficult to decompose. Lime is used to accelerate the breakdown process in the garbage. These components are mixed at a ratio of 5 kg (lime) per 1000 kg (plant materials) to produce high-quality compost. Lime mixed with water may result in the formation of a semi-solid substance or a dry powder. Lime boosts humification of plant wastes by decreasing lignin structure and improving humus content [40]. Likewise, usable compost substrates can be generated from various crop leftovers using a suitable process and quality control procedures (Table 1).

- Municipal solid waste (MSW)

Increasing population, industrialization, and urbanization has elevated the levels of MSW, which has become a problematic responsibility in Pakistan and worldwide [41].

The most well-known biodegradable waste procedures are microbiological stabilization and composting [42]. Due to the high organic content of MSW, composting is theoretically one of the most suitable AWM technologies for MSW management [43–45]. In addition, it generates a soil layer known as a conditioner with agronomic benefits, and is an economically viable and valuable method for offsetting the organic part of the trash. It also reduces the disposed waste, remarkably decreasing the residual waste's pollution capacity and volume for landfilling. As a result, numerous developing Asian countries are turning to compost to manage their MSW. Picking, contaminant separation, sizing and mixing, biological decomposition, and other functions are all part of the modern MSW management composting system. Figure 2 shows the schematic flow diagram of the distinct method of MSW management from source to utmost disposal. To weigh Pakistan's Lahore compost waste intake, a weighbridge having a capacity of 75 tons is located at the Mahmood Booti

open dumping site operated by the City District Government of Lahore [46]. Composting is primarily a small-scale industry in Bangladesh and the Maldives. MSW composting in Indore and a large-scale aerobic device in Mumbai were installed in India in 1994 to control 500 metric tons of MSW [44]. These are the two examples of operational large-scale composting ingenuities in India [47]. By 2008, composting had been used to treat 9% of India's MSW [44]. The average cost ranged from $25 to $30 per ton, while the market value per metric ton ranged from $33.5 to $42. India intends to add other plants in near future [48].

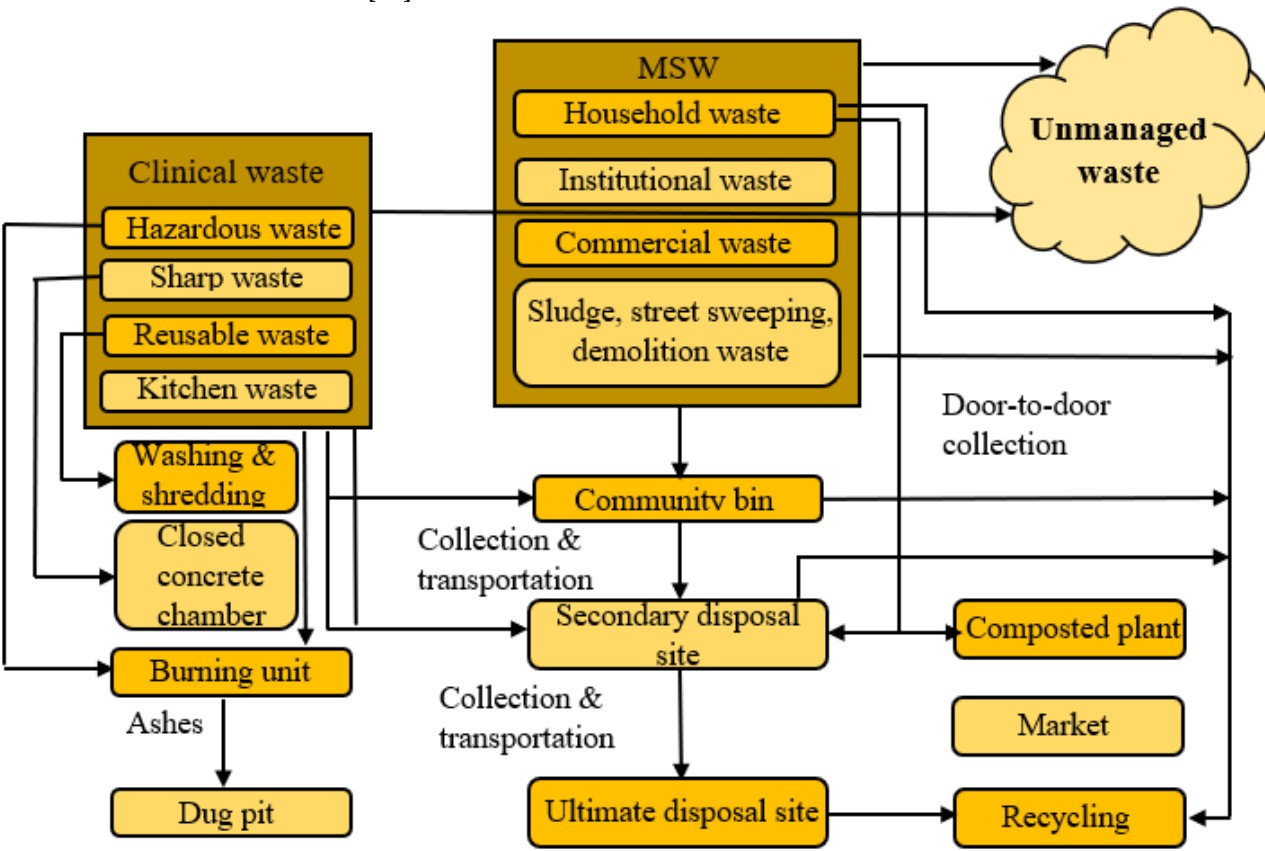

**Figure 2.** Schematic flow diagram of MSW management by composting.

- Biomedical waste (BMW)

The waste produced through the diagnosis, immunization, and treatment of human beings, research practices, and animal is organic BMW. In Pakistan, hospitals make approximately 2.07 kg of BMW per bed per day [49]. If BMW is not managed properly, it may cause serious environmental and health issues [50]. Therefore, safe disposal techniques need to be investigated, and composting is a sustainable option. Neem and tobacco extracts are commercially cost-effective for local small farmers and provide the best degradation of organic BMW. Thus, these extracts can be employed for conversion of organic BMW into potential fertilizer [51]. Previous research revealed that the BMW must be similarly treated with 5% sodium hypochlorite (NaOCl) at the disposal location [52]. It can be exposed to an initial decomposition process by mixing it with cow dung slurry, and then VC can be utilized to treat it further. Several epigeal species of worms may be used for this purpose. By using this approach to handle BMWs, these worms are more effective in decomposition. VC and proper handling of BMW can be energy-efficient and sustainable methods of eliminating and recycling this hazardous waste [52]. Meanwhile, the composting processes of various wastes come in discrete modes. The most utilized techniques are conventional composting, i.e., AC, AnC, and VC, and emerging composting, e.g., two-stage composting, as described below.

**Table 1.** Treatment methodologies of different types of crop residues.

| Waste | Physicochemical Characteristics | Method | Quality Control | Final Products and Uses | Outcomes | References |
|---|---|---|---|---|---|---|
| Barley waste | Composting in an open-air pile that was rotated 7 times in 105 d. Average temperatures of 65–68 °C with relative humidity of 45–65%. | Maximum temperatures of 65–68 °C with humidity of 45–65%. | Composting | Fertilizer | Micronutrient absorption favored at lowest doses. Doses >10 mg/L inhibited it and depressed growth at highest levels. | [53] |
| Barley straw waste | Conductance (compost to water, *w/w*: 1:3). pH (in water and 0.01 M CaCl) Quality of dry matter (% fw, 105 °C) Ash content (% dw, 480 °C/16 h) in triplicate. | Heterotrophic mesophilic bacteria. | Composting | Composting of cow and swine waste with barley straw. | 1—C/N ratio declined from 22.6–28.5 to 12.7 during composting. 2—Approximately 11–27% and 13–23% of total C and N were lost after 7 d of intensive composting and 62–66% and 23–37% for whole composting, respectively. | [54] |
| Barley waste | Final compost pH was 8.7 and C/N ratio was 13. No. of seeds germinated in co-compost depending on grains used. | Total OM was estimated by weight loss on ignition at 540 °C/16 h, and moisture on drying at 105 °C/24 h. | 1—Composting 2—VverC | OM composition was high in barley wastes and solid poultry manure. | OM content of barley waste was high (86.3% dw) and had N deficiency. | [55] |
| Wheat straw waste | Compost contributed 10% of its total N for plant growth during growing season. | During growing season, compost supplied 10% of available N to plants. | 1—Mature composting 2—Immature composting | Additional fertilizer | 1—At 126.5 h, total H yield of 68.1 mL H/g TVS was 136-fold higher than raw wheat straw wastes. 2—Substrate pretreatment was essential in turning wheat straw wastes into biohydrogen by composts producing hydrogen. | [56] |
| Rice straw | Lowest C/N ratios found (17–24). Pathogenic micro-organisms were extracted from rice straw by heating at 62 °C/48 h. | Micro-organisms respiration behavior was determined on separate initial C/N (17, 24, and 40) raw materials. | Composting | Development of paper, building materials, soil incorporation, manure, energy supply, and animal feed. | Rice straw residues was rich in OM (80%), oxidizable organic C (34%), and C/N ratio (very volatile and average of 50), suggesting a potential C supply for micro-organisms that can tolerate composting conditions. | [57] |
| Wheat straw waste | Overall C and N of materials was estimated. Wheat straw has C/N ratio of 100 and cover-grass hay has C/N ratio of 15. | Weight loss of compost samples oven-dried at 80 °C/24 h to assess water content. | 1—C1- Automatic NC analyzer connected to isotope mass spectrometer measured total N and C. 2—$NH_4$ and NO analysis—Traditional calorimetric approaches of flow-injection analysis. | Fertilizer | 1—pH ranged 7.6–8.9, with highest values after 3–4 weeks. 2—Weight loss after weeks of composting reduced by 44–45% of original weight. 3—After 7 1/2 weeks, weight loss was 61–63% of actual weight. 4—4% N rose from 2.8 to 4.6%. | [58] |

**Table 1.** *Cont.*

| Waste | Physicochemical Characteristics | Method | Quality Control | Final Products and Uses | Outcomes | References |
|---|---|---|---|---|---|---|
| Wheat straw waste | pH = 6.9 Negligible CaCO$_3$ content Organic C content of 11.0 g C/kg dry soil | Three types of UWC were applied 1—Bio-waste compost (BIO) from green waste and source-separated organic fraction. 2—Co-compost from mixture of 70% green waste and 30% sewage sludge. 3—Municipal solid waste compost. | 1—CERES model 2—Parameter modelling | Soil conditioner or fertilizer | 1—Simulated N fluxes indicated that organic amendments resulted in additional leaching of up to 8 kg N/ha/year. 2—After many years, composts mineralized 3–8% of their original organic N content. Composts with slower N release delivered more N to crops. 3—CERES used to help choose best time to apply compost. | [59] |
| Rice flakes | pH = 7 | Aspergillus spp. | Composting | Edible products | 1—As opposed to inorganic N, organic N contributed to higher enzyme production. 2—Optimum enzymatic activity was observed at 55 °C/pH 5. 3—Presence of Ca increased enzyme activity, while EDTA presence had opposite effect. | [60] |
| Rice straw | Temp., air circulation, moisture, and nutrients should all be appropriately managed. Initial optimal composting ratio of C/N was 25–30. | Psychrophilic and mesophilic micro-organisms. | AnC | Combination of swine manure and rice straw as fertilizer. | 1—Organic compound biodegradation caused temperature increase to 40–50 °C. 2—pH in all composts were constant and steady. | [61] |
| Rice straw | Gravimetric approach to assess moisture content. In-house approach was used to evaluate P and K amounts. | Composting in shaded environment on premium Agro products premises. Two therapies: compost piles with EM (C1) and without EM (C2). | Composting | Final compost in matured stage range could be used without limitation. | Compost treated with EM produces more N, P, and K (*P 0.05*) than compost without EM treatment. | [62] |
| Rice straw | Individually homogenized substrates and inoculum were deposited at 4 °C for further use. | Effect of characteristics on bio gasification was calculated using Box–Behnken experimental design combined with response surface methodology. | AnC | Research contributes to understanding of intertwined symptoms and microbial activity of Alzheimer's disease. | Bio-gasification of SS-AD of composting RS had significant interactive impact on temperature, ISC, and C/N ratio. Highest biogas output achieved at 35.6 °C with 20% ISC and 29.6:1 C/N ratio | [63] |

**Table 2.** Physiographical properties of organic feedstock materials or different wastes.

| Properties | Total Organic C (g/kg) | Total N (g/kg) | C/N ratio | pH | Total P (g/kg) | Total K (g/kg) | Reference |
|---|---|---|---|---|---|---|---|
| Household waste | 368 | 21.7 | 17 | 4.9 | | | [64] |
| Manure | 330 | 22 | 15 | 9.4 | 3.9 | 23.2 | [65] |
| Wood chips | 394 | 14.3 | 28 | 7.4 | 3.5 | | [66] |
| Sawdust | 490 | 1.1 | 446 | 5.2 | 0.1 | 0.4 | [65] |
| Canola | 457 | 1.9 | 24 | 6.3 | 1.1 | - | [67] |
| Rice | 412 | 8.7 | 47 | 6.8 | 1.1 | - | [67] |
| Soybean | 440 | 23.8 | 18 | 6.3 | 0.9 | - | [67] |
| Pea | 436 | 35.0 | 12 | 6.3 | 4.6 | - | [67] |
| Rice straw | 39.20 [1] | 0.64 [1] | 61.3 | 7.6 | 0.21 [1] | 1.12 [1] | [62] |
| Rape straw | | 6.52 | 59.8 | 7.11 | 0.99 | 31.64 | [68] |
| Wheat chaff | | 5.24 | 73.8 | 6.93 | 0.62 | 19 | [68] |
| Maize chaff | | 9.41 | 46.5 | 7.03 | 0.93 | 22.93 | [68] |
| Rice chaff | | 8.51 | 49.1 | 7.82 | 0.88 | 25.31 | [68] |
| Wheat straw biochar | - | 1.38 [1] | 38 | 7.03 | 0.45 [1] | 1.06[1] | [69] |

[1] Values in percentage. Total N = Total concentration of N. Total P = Total concentration of P. Total k = Total concentration of K.

## 3. Conventional Composting

In the recent few decades, conventional composting processes, i.e., VC, AC, and AnC, have become commonly used globally. Many investigations illustrated that AWM utilization in the field in the form of composting enhanced the soil texture, and structure and has many other beneficial impacts on the field. Researchers focused on improving the composting structure, providing bioavailable components, enhancing the product consistency, and the economic and environmental effects due to the advancement of approaches and green development. This section outlines the overall features of traditional composting and the resulting compost consistency.

### 3.1. Vermicomposting (VC)

VC is defined as utilizing organic waste from several earthworm species [70–72], and it occurs in a bin/tub. A bin is prepared with a perforated bottom made of adjacent layers of 0.5 mm and 1 cm sieve sizes of nylon and aluminum to facilitate compost tea infiltration. For instance, cow manure was placed at the bottom with worms, including Eudrillus Eugineae/Eisenia Fetida, on top, and shredded kitchen waste was placed over the worms. One bucket of water was added daily for the survival and multiplication of worms. Water, when it leaches down, can be used as compost tea. Compost tea is a liquid fertilizer enriched in nutrients that can be applied for plant growth enhancement. The VC procedures are illustrated in the schematic diagram in Figure 3.

Several different species of worms have been used with different combinations of organic materials (waste) with the purpose of their degradation and conversion into a value-added product. Table 3 shows discrete composting parameters utilizing VC. Earthworms can decay various types of OM, including sewage sludge [73–75], cattle farm waste [76,77], poultry waste [78,79], bagasse [80], industrial waste [81–83], and residential waste [84]. Sludge is the most widely studied for VC, followed by household waste (Table 3). Worms, including Eisenia Fetida, Eudrillus eugineae, Perionyx Sansibaricus, Pontoscolex Corethrurus, Megascolex Chinensis, and Lampito Mauritii are quite effective for VC.

Comparison of compost production from organic waste between different earthworm species, including Eisenia Fetida vs. Lampito Mauritii and Eisenia Fetida vs. Eudrillus eugineae, is shown in Table 4. This study was conducted in a semi-arid climate in Jodhpur, India. The amount of N, P, and K increased while the amount of C/N and C/P decreased as VC preceded. The ideal temperature, moisture content, and pH of Eisenia Fetida were 25 °C, 75%, and 6.5, respectively, for optimum growth, while those of Lampito Mauritii were 30 °C, 60%, and 7.5, respectively. For optimum growth of the earthworm species, ideal temperature, moisture content, and pH were 25 °C, 75%, 6.5 for Eisena Fetida and 30 °C, 60%, and 7.5 for Lampito Mauritti, respectively. The results showed that Eisenia Fetida

produced nutrient-rich compost more effectively and efficiently than Lampito Mauritii [85]. Performance evaluation of Eisena Fetida was accessed for six different poultry waste combinations, cow dung, and food industry waste in the semi-arid climate of Hirsa, India, and the results showed an increase in N, P, and K and a decrease in the C/N ratio. Eisena Fetida performed best when cow dung was mixed with poultry waste and food industry waste in a ratio of 2:1:1 compared to cow dung alone [86]. In another study in Kolkata, India, a combination of Eisenia Fetida with micro-organisms, including N-fixing, K-fixing, and P-solubilizing bacteria, was utilized for compost formation using sawdust, paddy straw, and water hyacinth as compost feedstocks. The results showed that not only the time for compost production was reduced, but the percentage of nutrients was also increased in the final product (Table 4). Paddy straw and water hyacinth provided better results than sawdust in compost formation [87]. Comparative studies between Eisenia Fetida vs. Eudrillus eugineae in compost production revealed that in 100 gm compost, 250 worms of the local species, Eisenia Fetida, or Eudrillus eugineae yielded 7, 11, and 17 cocoons and 460, 227, and 540 juveniles per 100 gm, respectively. Around a 40-fold increase in Eudrillus eugineae was achieved, while there was only a 10-fold increase in the local earthworms. Eudrillus eugineae produced compost within 40 days, while local species took 50 days to prepare the final compost [88,89]. Combinations of earthworms with micro-organisms (N-fixing, K-fixing, and P-solubilizing bacteria) minimized the time duration for composting and the finished compost was more highly enriched in nutrients (N, P, K, Ca, Mg, Zn) than conventional compost. As reported, adding P-solubilizing, N-fixing, and K-fixing bacteria increased the amount of N, P, and K in the final compost [90].

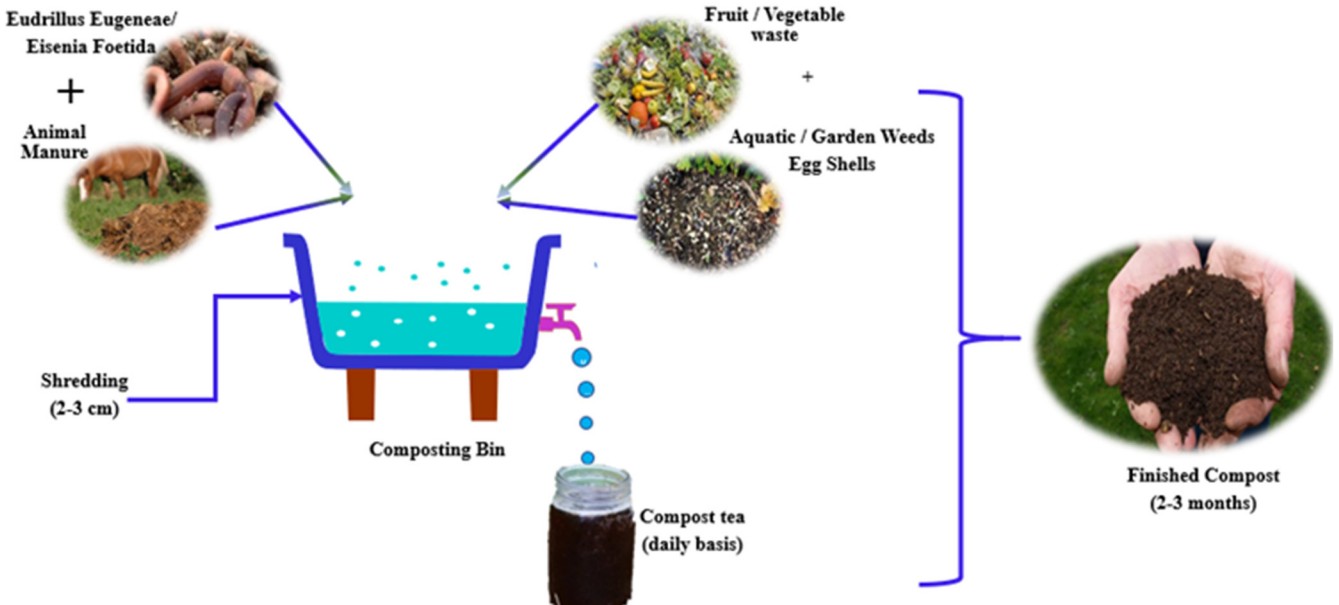

**Figure 3.** Schematic of vermicomposting production process.

In Kerala, India, banana, cassava, and cowpea composting materials were inoculated with N-fixing, K-fixing, and P-solubilizing bacteria. The results showed that Eudrillus eugineae performed better than other earthworms in synthesizing the nutrient-enriched compost. Using prepared Vermicompost as fertilizer provided the best results in accelerating root growth by increasing nutrient uptake and enhancing total yield [89]. Peppermill sludge, solid pulp, and cow dung were fed to Eudrillus eugeneae to reduce pollution and convert waste into value-added products. The earthworms survived and resulted in enhanced N and P content and a reduction in the C and N ratio, demonstrating the efficiency and effectiveness of Eudrillus eugeneae [91]. Four different feedstocks, namely seaweed, sugarcane trash, coir pith, and vegetable waste, were used for composting with Eudrillus eugineae. Composting for 50 days revealed that different parameters, including pH, OMC,

TOC, N, P, K, cellulose, and lignin, were decreased compared with the C/N ratio. An upsurge in the nutrients in vermicompost showed its development. It was concluded that if cow dung was added to a mixture of materials, it would enrich the nutrients in the final compost. The reproduction and growth rate of Eudrillus eugineae increased as the amount of C/N ratio increased [88]. Eudrillus eugineae was the best in forming nutrient-enriched products in less time and at a greater reproductivity rate than Eisenia fetida and local worms. Different combinations of worms, bacteria, and organic waste generated higher quality vermicompost when utilized together.

**Table 3.** Discrete composting parameters utilized in the vermicomposting process.

| Type of Waste | Factor | Range | References |
| --- | --- | --- | --- |
| Sludge from Tannery and cattle dung | C/N ratio | 19.00 | [92] |
| Cattle dung and tannery sludge | pH | 9.02 | [92] |
| Newspaper and sawdust | pH | 7.23 | [93] |
| Distillery industry sludge | pH | 6.70 | [94] |
| Distillery industry sludge | C/N ratio | 19.50 | [94] |
| Household waste | pH | 7.43 | [94] |
| Household waste | C/N ratio | 9.89 | [94] |
| Sludge from WWT plants | EC (mS/cm) | 1.81 | [74] |
| Wastewater treatment plant's sludge | pH | 6.9 | [74] |
| Mixed (farmyard manure, agriculture, and MSW) | C/N | 18.6 | [95] |

**Table 4.** Chemical properties of different worms in vermicomposting.

| Worms | Composting Materials | N | P | K | C/N | C/P | Reference |
| --- | --- | --- | --- | --- | --- | --- | --- |
| Eisenia Fetida Lampito Mauritii | Sawdust Straw Biogas slurry Cow waste Kitchen waste | 3.32-fold increase | 1.61-fold increase | 1.13-fold increase | 2.79-fold decrease | 1.35-fold decrease | [85–87] |
| Eisenia Fetida | Cow waste Poultry waste Food waste | 1.6–3.6-fold increase | 33.7% −54% increase | 39.5% −50% increase | 10.7–12.7 decrease | N/A | [85–87] |
| Eisenia Fetida Trichoderma viride (M) Bacillus polymixa (M) Azotobacter Bacillus firmus (M) chroococcum(M) | Water hyacinth Paddy straw Sawdust Food waste | 52–72% increase | 34–80% increase | 45–80% increase | lowest from initial | N/A | [87,89] |
| Eisenia Fetida Eudrillus eugineae (B) Perionyx sansibaricus Pontoscolex corethrurus Megascolex chinensis | Banana Cassava Cowpea | 62% increase | 20% increase | 38% increase | 11 points | N/A | [87,89] |
| Eudrillus eugineae | Solid pulp Paper sludge Cow dung | 63.31% increase | 2–11-fold increase | N/A | 9.6 points | N/A | [89,91] |
| Eudrillus eugineae | Seaweed Sugarcane trash Coir pith Vegetable waste | 63.75% increase | 31.58% increase | 42.55% increase | 23.91 for seaweed | 46.04 for seaweed | [88] |

## 3.2. Aerobic Composting (AC)

AC is the degradation of OM with micro-organisms by utilizing oxygen and it takes place in the open atmosphere as a pile or pit [96–99]. For instance, green and brown materials are shredded by a chopper and to a size of 2–3 cm or smaller to help in rapid decomposition. The shredded material is then arranged in a pile/windrow with a specific moisture content. Frequent turnings are employed with sufficient moisture for proper mixing and provide aeration to ensure micro-organisms' survival. The micro-organisms multiply in organic material with sufficient water and air and decompose organic material. After seven to eight turnings, the material becomes fine and changes its color to dark brown (depending upon the material used for composting) with reduced odor. Now the compost

is ready to use as organic fertilizer. The schematic flow diagram of this process is shown in Figure 4.

**Figure 4.** Schematic flow diagram of AC.

A couple of windrows are developed, aerated and turned with an air pump, and mechanically turned by a tractor installed with a bucket loader. Compost prepared with both methods has the same characteristics and organization (60 days). Both composting processes have the same temporal changes in temperature, biological, physical, and chemical parameters, as exhibited in Table 5. However, thermophilic micro-organisms eliminated the harmful bacterium fecal coliform due to increasing temperature [100]. A mixture of poultry waste, feed waste, wood chips, and feathers was used in compost formation under aerated piles, and the effect of the produced compost on soil and crop production was assessed. The results showed that several changes occurred during compost formation, e.g., temperature rise due to mesophilic and thermophilic micro-organisms and a change in OM, N, P, and K levels even in piles that were aerated but not turned. Composted poultry litter had significantly more OM than un-composted poultry litter. Thus, with the availability of OM, crop fields are less susceptible to loss [101]. Another study was performed to determine the suitability of aerated and turned piles using olive husks as the compost material. The outcome showed that both piles reached their maturity stage simultaneously, while the thermophilic phase of turned piles was achieved earlier and had slightly higher OM than the aerated pile (94% versus 84%). The variations in chemical and biological parameters were negligible in both piles, as shown in Table 5. For large-scale applications, the mechanical turning method is best to convert waste into a valuable resource as higher temperatures are achieved through mechanical turning [102].

A comparative study was carried out between aerated and turned windrows to evaluate their effectiveness, using olive mill waste mixed with grape stalks and sheep litter as composting materials. Both methods evaluated efficiency based on pH, temperature, OM, and total N. The results showed that several drawbacks were associated with the aerated composting process due to the physical properties of olive husks. The prepared compost

from both methods had similar characteristics, while the thermophilic duration of the turned compost lasted longer and had higher humification than the aerated compost [103]. Different studies were conducted utilizing chicken manure, wheat straw, and bamboo biochar (Table 5). The physicochemical processes, biological parameters, and gas emissions were periodically measured to assess compost quality. Adding biochar enhanced porosity and stabilized the composting rate, accelerating the process and improving the finished compost's quality. Biochar improves looseness, provides better material degradation, and reduces GHG emissions ($CO_2$, $N_2O$, $NH_3$, $CH_4$) [104]. The composting experiment was performed in two bins with different composting materials, and the effects of low C/N ratio on the final product, including several parameters are shown in Table 6. Less straw and more swine manure were added into bin one, which had a low thermophilic duration and took longer to mature than bin two which had more straw and less swine manure. It was recommended that 172 kg of straw could be treated with one ton of swine manure. A low C/N ratio was recommended for composting rice straw with swine manure [61]. In-vessel composting offers fewer complications than windrow- or pile-composting due to reduced bioaerosols, and better AWM and control over leachate.

The composting material consisted of three different types of waste: green waste, paper waste, and bio-solids in bins 1, 2, and 3, respectively. Better results were attained from bin 3 with maximum temperature achieved and a more humified final product (Table 6). The active compost was carried out in bins, whereas the compost was taken out and matured by successive turnings to complete the maturity phase. However, precautions must be taken during the maturity phase to reduce pollution and effective AWM [105]. Several sleeves were used to prepare the compost with an equal ratio of green waste and sewage sludge. The concentration of oxygen was maintained through a perforated pipe, which was inserted at the bottom of the sleeve. Controlled moisture content and thermophilic temperature were maintained at >45 °C throughout the composting. Harmful bacteria, including Fecal coliform and E. coli, dominated initially but were subsequentially reduced, and the traces of these micro-organisms in the final product were negligible. The active phase of composting was performed in the sleeves while the maturity phase was carried out in the open. The final product was non-toxic and used as a beneficial soil additive [106].

Composting in sleeves results in reduced odor and attracted fewer insects with better leachate management compared to open windrows/piles. The composting materials included green waste and olive mill wastewater, in which the green waste remained soaked for one night. The oxygen level was maintained by the addition of a perforated PVC pipe through which air was injected into the sleeve. The temperature throughout the entire sleeve was maintained at >45 °C (Table 6). The final compost had no toxicity and basil and ornamental plant growth were tested using prepared compost. In this way, wastewater was beneficially utilized and converted into a valuable resource [107]. In another study, the performance of a closed bioreactor (In-vessel) was evaluated in terms of various physicochemical parameters, including C/N ratio, $NH_3$-N, pH, moisture, N content, etc. Mixtures of different food wastes were collected from several locations and placed in a bioreactor with the recommended initial standards. The final compost was ready to use within 12 days. During composting, temperature, $CO_2$ levels, and pressure rose due to microbial activity, resulting in satisfactory final compost that was acceptable for agricultural applications. Nitrates negatively correlated with $CO_2$, EC, and ammonium levels, while phosphate positively correlated with ammonium, EC, and $CO_2$ levels [108].

AC has already been tested under different aeration methods using cotton gin waste [109], straw and sheep manure [110], poultry litter [111], and sawdust [100], and literature reveals that various methods have been used for AC in the past. Windrow and forced-air composting are of more significant concern as both yield similar results. Energy and cost consumption in forced air composting is higher than in windrow composting. Perforated PVC pipes were laid under waste for air circulation through air pumps to aid in waste degradation. The temperature achieved by thermophilic and mesophilic organisms was also lower in forced air composting than in windrow composting. Higher temperatures are necessary for the

elimination of harmful pathogens from waste. Some studies reveal that higher temperatures can be achieved in windrow composting but controlling this temperature is complicated by the frequent turnings. Manual turning involves intensive labor application as compared to mechanical turning. On the other hand, mechanical turning saves time and reduces cost as compared to manual turning. The degree of humification in mechanically or manually turned windrows (compost) is far greater than in forced aerated (air pumping) compost. Because of rapid evaporation, the moisture content requirement of forced aerated compost is much higher than turned composting. The amount of ideal OM and particle size of compost is higher in manually or mechanically turned end-stage compost than in forced aerated compost because no turning is carried out in aerated compost. According to studies on mechanically or manually turned AC compost, it is superior in all aspects compared to forced aerated composting.

**Table 5.** Chemical properties of different materials in various aeration methods.

| Aeration Method | Materials | Temperature Achieved (°C) | pH | Total N | Total P | Total K | C/N Ratio | Reference |
|---|---|---|---|---|---|---|---|---|
| Turning | Pig waste Sawdust | 67 | 6.7 | 18–27 g/kg | N/A | N/A | N/A | [100] |
| Air pump | Pig waste Saw dust | 60 | 6.8 | 18–27 g/kg | N/A | N/A | N/A | |
| Air blower | Poultry manure Wood shaving Waste feed Feathers | 63 | 7.0 | 16.31 g/kg | 15.57 g/kg | 19.78 g/kg | 15 | [101] |
| Mechanical turning | Olive oil husk Grape stalks | 65 | 7.3 | 0.95–1.17% | N/A | N/A | 46 | [102] |
| Centrifugal ventilator | Olive oil husk Grape stalks | 54 | 7.1 | 1.08–1.27% | N/A | N/A | 46 | |
| Forced aeration | Waste from olive mill Sheep litter Grape stalks | 63 | 9.5 | 1.94% | 0.9 g/kg 2.5 g/kg 1.3 g/kg | 2.4 g/kg 2.7 g/kg 2.8 g/kg | 15.5 | [103] |
| Windrow turning | Waste from olive mill Sheep litter Grape stalks | 68 | 9.2 | 1.89% | 0.9 g/kg 2.5 g/kg 1.3 g/kg | 2.4 g/kg 2.7 g/kg 2.8 g/kg | 15 | |
| Centrifugal ventilator | Poultry manure Bamboo biochar Wheat straw | 55 | 9.0 | 57% at the final stage | | | 10.3 | [104] |

**Table 6.** Physicochemical characteristics of different materials in vessels.

| Vessel | Materials | Temperature Achieved (°C) | Total N | C/N Ratio (%) | pH | Moisture Content (%) | Reference |
|---|---|---|---|---|---|---|---|
| Bin-1 | Swine waste Rice straw | 60 | 19.30 g/kg | 5.15 decrease | 8.01 | 45 to 65 | [61] |
| Bin-2 | Swine waste Rice straw | 60 | 18.62 g/kg | 4.57 decrease | 8.03 | 45 to 65 | |
| Bin-1 | Green waste | 64 | 200 mg/kg | 20 ± 1 | 8.5 | 81 ± 33 | |
| Bin-2 | Green waste Paper waste | 70 | 120 mg/kg | 25 ± 1 | 8.5 | 72 ± 2 | [105] |
| Bin-3 | Green waste Biosolids | 72 | 700 mg/kg | 27 ± 3 | 8.9 | 44 ± 11 | |
| Sleeve-1 | Green waste Sewage sludge | >45 | 44.9% loss | 10.9 decrease | 6.5 at sleeve opening | 54.4 | [106] |
| Sleeve-2 | Green waste Sewage sludge | >45 | 42.9% loss | 11.8 decrease | 7.2 at sleeve opening | 38 | |
| Sleeve | Olive mill waste Green waste | 55 | 1.05% | 21.5 | 8.2 at sleeve opening | 55 | [107] |
| Vessel | Mixed food waste | 53 | 250 mg/kg | 11 | 6.94 | 72 | [108] |

### 3.3. Anaerobic Composting (AnC)

AnC degrades OM in the presence of micro-organisms without oxygen utilization [112–114]. AnC occurs in two steps. For instance, cow dung was fed daily in a digester for biogas generation to utilized in the first step. A byproduct of the digester was slurry, referred to as digested, from which all GHGs were eliminated. The product was utilized effectively for composting and reduced environmental pollution. The slurry was mixed with shredded browns enough moisture to carry out further decomposition. The period of AnC is comparatively more extended than AC, and the schematic diagram is shown in Figure 5.

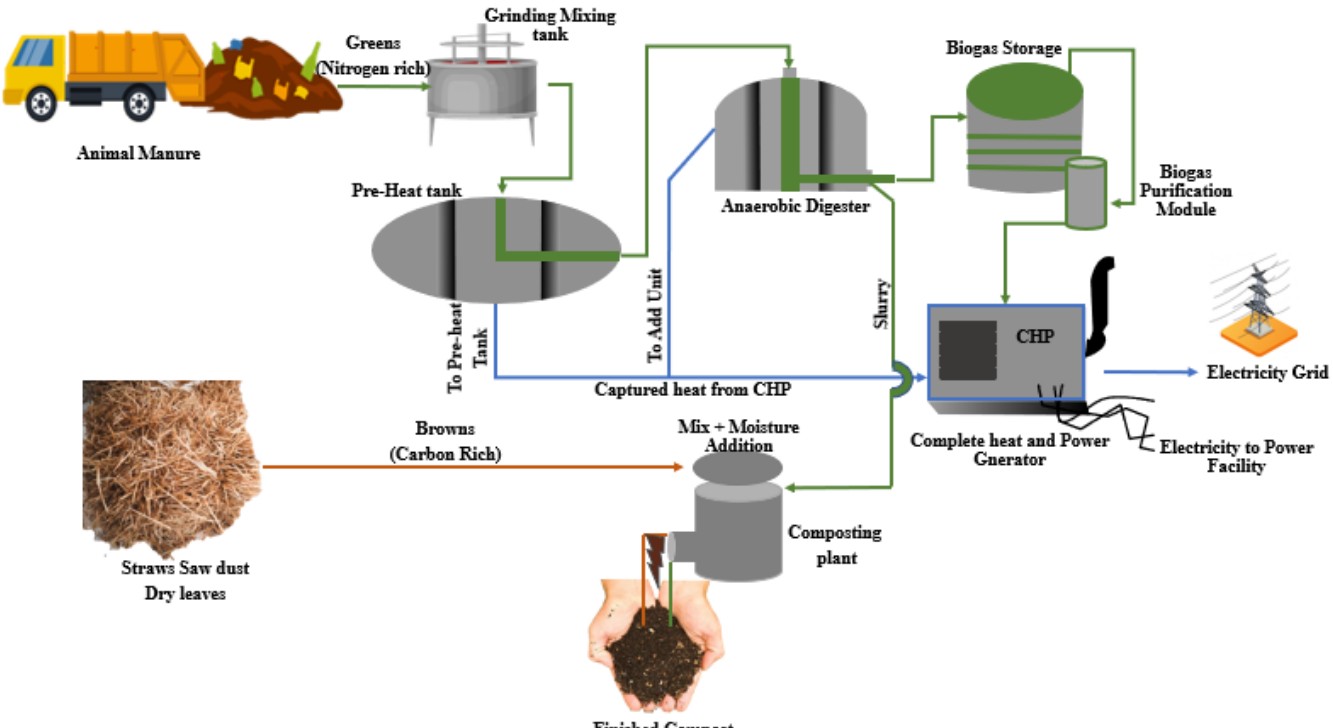

**Figure 5.** Schematic diagram of AC.

In the past, AnC has been used for the degradation of kitchen waste, fly ash and crop deposits [115], garden and animal waste [116], municipal waste [117], sawdust and pig manure [118], and sewage sludge [119]. When AnC was utilized for the above-mentioned feedstocks, it reduced GHG emissions since it occurs in an oxygen-free environment [120]. GHGs are regarded as harmful gases contributing to global warming, eutrophication, and acidification if high amount enter the atmosphere [121]. Methane produced by anaerobic digestion can be exploited as an energy source for either electricity production or combustion. Anaerobic digestion prevents environmental pollution as the generated methane is used and thus gets removed by burning. AnC requires a high amount of moisture and N-enriched material (animal manure, food waste, and sewage sludge) for the successful completion and generation of beneficial end-user products. Due to the high moisture at the end of AnC, compost tea is produced that can be used as liquid fertilizer that is enriched in nutrients (N, P, and K). In addition, the emission of volatile compounds (terpenes, ethers, and esters) during active composting periods in AnC is negligible [122]. The preparation of the final product from anaerobic compost requires a longer time. As a drawback, the final product of anaerobic digestion contains E. coli and Salmonella, which are hazardous to human health. Another negative impact of anaerobic digestion is the production of odors.

## 4. Emerging Composting

Two-stage composting is a technology that incorporates two diverse methods into a single composting method to improve the finished product quality, process speed, and environmental impact of traditional composting. It is a novel approach to bio-fertilizer use. Various two-stage techniques have been considered, including combining two composting technologies, e.g., VC and conventional composting. However, in this paper, AnC followed by AC is reviewed. AnC is termed primary composting (PC) in the two-stage composting process, and the AC process is termed secondary composting (SC).

AnC followed by AC is a comparatively innovative idea in two-stage composting. As an initial effort, [96] investigated the transformation of OM and the kinetics of sewage sludge composting in two stages using grass and rape straw. The whole procedure required 217 days, with 10 days in the bioreactor for OM oxidation and waste sanitation and 207 days in the windrow for compost maturation. The concept of two-stage composting is recent and research on its economic and environmental effects is minimal. Most of the studies are aimed at increasing process reliability and product consistency. Table 7 summarizes recent studies into two-stage composting and the results obtained using various additives. A steady higher temperature of about 70 °C was established in PC during two-stage composting [123]. Within 40 days of switching to SC, the temperature fell from 50 to 30 °C. There will be less N loss from SC at this temperature at the mesophilic stage.

Moreover, since a higher temperature was observed during PC, N loss and GHG emission in the bioreactor can be minimized or regulated within a minimal range. According to [124], more thermophilic phases were noted, with one occurring during the PC and two to four occurring during the SC. Most of the thermophilic steps that followed were over 55 °C. Bamboo vinegar was added to the compost throughout SC to decrease the hazard of N degradation.

The thermophilic temperature range was between 55 and 70 °C in all the studies mentioned in Table 7. The high temperatures completed the maturation of the pile, ensuring the compost's safe use as a bio-fertilizer. Similarly, various feedstocks, e.g., pig manure, poultry manure, and other supplemental waste substrates, can be used in co-composting operations [125,126]. When AC is replaced with two-stage composting, there is a reduction in processing time. The addition of a bulking agent, particle size reduction, and aeration rate change are all essential considerations in determining the process performance and the finished product's consistency [127]. On the other hand, AnC followed by AC will minimize the area, labor, and time required for AC, along with the capital cost and power depletion. Two-stage composting also reduces GHG pollution and waste conveyance costs if PC is controlled at waste occurrence locations to decrease the total of waste before transport to the location for SC. Two-stage composting may be a novel way to manage organic waste at home or market. The organic waste can be collected and composted in a digester near the city. The incompletely composted waste from several cities could be transported to AC sites for further treatment. AC alone has adverse effects on the environment and results in GHGs, including ammonia, methane, and nitrous oxide, which may cause ozone depletion and global warming. On the other hand, AnC eliminates harmful GHG emissions into the atmosphere and it is a great energy resource. This is consistent with the literature about two-stage composting (AnC followed by AC) because AnC cannot be utilized directly without further treatment. It is rich in ammonia content which could burn crops. AC overcomes this hazard and converts the byproducts into useful resources. Two-stage composting yields better results than aerobic, anaerobic, and VC in terms of humic substances, OM, energy generation (heat and electricity), environmental protection, and nutrient-enriched end products.

**Table 7.** Chemical properties of different materials.

| Materials | Final N | Temperature (°C) | CH$_4$ | C/N Ratio (%) | Final P | Final K | pH | EC | Reference |
|---|---|---|---|---|---|---|---|---|---|
| Rice straw | 0.78 | 35.6 | 346 mL g VS$^{-1}$ | 29.6 | N/A | N/A | N/A | N/A | [63] |
| Dairy manure Corn stover Tomato residue | 31.6 kg | N/A | 1186 gm | N/A | 47.4 kg | 279.1 kg | N/A | N/A | [121] |
| Kitchen waste Garden waste Paper waste | 60 g/ton | N/A | 100 m$^3$/ton | N/A | N/A | N/A | 8.6 | 1.8 | [122] |
| Banana Cow dung Poultry waste | 2.09% | 57 | N/A | 12.8 12.6 12.9 | 11.86% | 0.39% | >9.0 | 0.59 0.57 0.65 | [123] |
| Pig manure | 1045 mg/kg | N/A | 18.6 mL/day | 8.7 | N/A | N/A | 8.5 | N/A | [124] |
| Food waste Inoculum | N/A | 37 | 2.27 m$^3$m$^{-3}$/day | 8.7 | N/A | N/A | 7.1 | N/A | [125] |

## 5. Comparison

Bio-waste combinations, including kitchen, garden, and paper wastes, have been utilized for aerobic and two-stage composting (combined anaerobic/AC). AC was carried out under forced aerated conditions for twelve weeks and two-stage composting in which AnC (PC) continued without aeration for three weeks while AC (SC) continued for the last two weeks. The results showed that AC yielded 742 g/ton of explosive gases, while two-stage composting yielded 236 g/ton and 44 g/ton of volatile gases (esters, terpenes, ethers, compounds). Biogas produced during the anaerobic phase utilized in combustion resulted in 99% removal of combustible gases. Two-stage composting was an attractive method for reducing volatile gas emissions [122]. Another study of prepared of anaerobic compost of the paper industry and urban solid waste was assessed in total ammonia, germination indices, volatile organic acids, and total oxygen uptake. The results showed that the application of prepared anaerobic compost was less effective unless anaerobic composting was followed by AC, which yielded better results [96]. The efficiency of AC and AnC was evaluated using different combinations of banana peel [plain banana peel (B), inoculated banana peel mixed with cow dung (BC), and poultry litter (BP)]. It was suggested that the decomposition rate in AC is faster than AnC at this scale with increased N and K content as follows: BP > BC > B [124]. The effect of AnC prepared from pig waste was assessed in neutralizing high chlorine content in soil due to polychlorinated biphenyl. If the chlorine content was significantly higher than the limit, then di-chlorination would be inhibited. Soil-to-organic waste ratio was 2:3, the C to N ratio was 20. At a moisture content of 60%, di-chlorination was the highest at 1 mg/kg [123].

A study was conducted to investigate the adverse effects [acidification potential (AP), eutrophication potential (EP), and global warming potential (GWP)] of various organic waste treatments (dairy manure, corn stover, and tomato residue). All treatment techniques used anaerobic digestion followed by composting. The results showed that if AnC was used before composting, EP, GWP, and AP were reduced. If AC and composting were used alone, the harmful potential concentration increased in the ecosystem. If the farm was equipped to use anaerobic digestion, then followed with composting would be suitable for all life cycle impact categories [121]. The performance of MUSTAC (Multistep sequential batch two-phase AnC) was evaluated, and the processes involved including hydrolysis, acidification, post-treatment, and methane recovery. This process was utilized for treating inoculated food waste using AnC. MUSTAC and anaerobic digestion were assessed in terms of environmental constraints. MUSTAC yielded the best results in reducing volatile emissions with high methane conversion efficiency attained in a relatively short period (Table 8). The product obtained could be used for soil improvement. MUSTAC has proven to produce value-added products with high efficiency and reliability [124]. A study was performed to assess the suitability of increasing

biogas from composting rice straw with the effects of primary temperature, C/N ratio, and substrate on the finished product. Concentrations of lignin, cellulose, and hemicellulose in the rice straw were sufficiently degraded. The initial concentration of the parameters mentioned above significantly affected bio gasification, as mentioned in Table 8. Methano-bacteria, clostridia, and beta proteo-bacteria were the microbial communities included in anaerobic compost, providing valuable information about microbial behavior and the independent effects of anaerobic digestion [63].

**Table 8.** Recent developments in two-stage composting.

| Time (d) | Waste | Composting System | Amendment | Remark Outcome | Reference |
|---|---|---|---|---|---|
| PC:1 SC: 207 | Dewatered sewage sludge | 1—Aerated bioreactor (PC: 1 m$^3$) 2—Weekly turned Windrow (SC: 0.8 m$^3$) | Different proportions of rape straw and grass. | 1—Feedstock composition affects process succession. 2—Rape straw raised temperature of compost and formation of humic acid. | [125] |
| PC: 6 SC: 24 | Green waste | 1—PC: non-covered digester, automated turning, and watering (daily) 2—SC: Windrow, turned and saturated every 30 d. | Brown sugar and calcium superphosphate in various proportions. | Proposed two-stage composting produce higher-quality compost in limited time. Adding 0.5% brown sugar and 6% calcium superphosphate to compost during SC increased consistency. | [126] |
| PC: 6 SC: 24 | Green waste | 1—PC: non-covered digester, automated turning, and (daily) 2—SC: windrow, turn, and water every 3 d. | Different proportions of rhamnolipid (RL) and initial compost particle size (IPS). | 1—Addition of 0.15% RL and particle size of 15 mm IPS increased aeration and water permeability, resulting in higher micro-organism numbers and enzyme activities, thus speeding up degradation process. 2—Mature compost of greater efficiency accomplished in just 24 d. | [126] |
| PC: 10 SC: 170 | Dewatered sewage sludge | 1—Aerated bioreactor (PC: 1 m$^3$) 2—Weekly turned windrow (SC: 0.8 m$^3$) | Aeration rate in bioreactor (0.5 and 1.0 L/min kg dm) was changed. | 1—A greater aeration rate in bioreactor resulted in OM losses. 2—Compost was safe to use as soil amendment because results exhibited low levels of heavy metals, low possible environmental risk, and suitable sanitary consistency. | [128] |

## 6. Conclusions and Recommendations

This review examines the management of AWM through various composting processes—conventional and emerging composting—and composting stages, the composting of crop residue waste, MSW, and BMW, as well as the underlying mechanisms, and the factors influencing composting. In addition, it compares conventional composting [vermicomposting (VC), aerobic (AC), and anaerobic (AnC)] with new composting techniques (two-stage composting). AW must be treated quickly and effectively for the sustainable growth of agriculture and environmental habitats. There are numerous ways to make valuable products from this massive volume of waste, but some are more cost-effective and/or rational. Composting is the most cost-effective and environment-friendly AWM practice, preferrable to landfilling, burning, and open-dumping of agricultural and farm wastes. Composting is crucial for recycling waste into resources, preserving environmental quality, and safeguarding public health. These methods include the recycling of AWM to increase soil fertility and the production of biofertilizers through different processes. The summary of composting phases and critical waste substrates demonstrates that composting is the most effective method for AWM. The literature reveals some conventional and emerging composting processes. In conventional composting, VC humifies in 3 to 4 months which cannot fulfill fertilizer demand. AnC also

requires 2 to 3 months to prepare for daily biogas and slurry production. Rapid compost preparation with a fine and higher degree of humification can only be done through AC. In this study, several past studies examine recent developments in organic composting. Several recommendations were made to improve technical development. This critical review also highlights current advancements in composting for AWM, makes recommendations to aid its technological development and acknowledge its benefits, and will boost the scientific community's interest in composting processes.

BSF larvae are best tool for AWM, which can also decompose AW quickly. The increased lignocellulosic component of AWMs limits their decomposition. Comparatively, VC with BSF reduces GHG emissions 47-fold. Future research should focus on discovering the ideal settings for BSF larvae to evolve, flourish, and handle MSW and crop wastes in subtropical regions. BSF can be employed to decompose recalcitrant AWM substrates. Centipedes and pill bugs are sometimes utilized in composting. Composting with these insects would help these plants and environments survive. They could be composted instead of discarded. Insects should be examined to see whether they can aid in macro- or micronutrient enrichment of compost. Compost can release pathogen-killing enzymes. As composts include several nutrients, they avoid providing mono nutrients. Before planting, a soil analysis can reveal nutrient deficiencies. Mono fertilizers extracted from compost fertilizer would reduce nutrient waste.

Farmers can use inoculum that degrades complex biodegradables to speed up the composting process. More study is needed to identify an odor-trapping technique to overcome the compost processing-related air quality issue. Implementing composting and $CO_2$ capture should reduce GHG emissions. AnC or other composting processes could use an odor-trapping device. Researchers and businesses have long recognized the waste potential as a source of raw materials. The chemical components of waste are of particular concern. In the past decade, a shift has been made from composting organic fractions for crop production to anaerobic digestion, which can produce methane as an energy source. Due to government incentives, European waste firms have changed their investments to anaerobic digestion systems. These government incentives may encourage new composting innovations, e.g., incorporating bioenergy technologies (anaerobic digestion, biochar). Bioenergy byproducts may be composted to maximize their economic, agricultural, and environmental value. To make compost more acceptable, anti-nematodes, viricides, bactericides, and fungicides generated from plants may also be added. By avoiding pesticides, organic farming would continue to be promoted. Slow-decomposing materials can be composted separately from other materials so that the composting period of the latter is not prolonged. There is a need for additional research to discover whether substances that require longer decomposition times tend to mineralize over time. Slow mineralizing minerals that serve as a long-term source of nutrients could be advantageous to biennial and perennial plants; this theory's validity should be studied further. This research could reveal the nutritional benefits of leaves that decompose slowly and will help determine whether they should be composted. Composting rather than burning agricultural waste is garnering more attention in developing nations. Before spreading compost onto the soil, it should be frequently evaluated for maturity and pollutants to prevent introducing potential hazards to the soil and other living things. Finally, additional trials are required to determine how to accelerate the composting process. Even though the two-stage composting process developed in the past continues to be an emerging composting process, the best practices will aid in sustaining the composting process.

**Author Contributions:** Conceptualization, U.W.H. and S.H.; methodology, S.H., M.W., R.N. and S.A.; formal analysis, M.W.; investigation, A.N., U.W.H., P.T.H. and M.W.; resources, S.H., M.S. and U.W.H.; data curation, M.W., A.N. and S.A.; writing—original draft preparation, S.H., S.A., H.A.L. and M.W.; supervision, U.W.H., M.S. and S.H.; project administration, U.W.H., S.H. and M.S.; funding acquisition, S.H., R.N. and U.W.H.; writing—review and editing, S.A., H.A.L., R.N. and A.N. All authors have read and agreed to the published version of the manuscript.

**Funding:** This research received no external funding.

**Data Availability Statement:** Data used to support the study's findings can be obtained from the corresponding author upon request.

**Acknowledgments:** The authors would like to express their gratitude to The Joint Graduate School of Energy and Environment (JGSEE); King Mongkut's University of Technology Thonburi; the Center of Excellence on Energy Technology and Environment (CEE); the Ministry of Higher Education, Science (MHESI) Research and Innovation and Department of Mathematics; and Muhammad Nawaz Shareef of the University of Agriculture, Multan, Pakistan for their support and technical help. We are thankful to Muhammad Faheem from the Department of Environmental Science and Engineering, School of Environmental Studies, China University of Geosciences for English editing services.

**Conflicts of Interest:** The authors declare no conflict of interest.

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
