# Peer review of "Composting Processes for Agricultural Waste Management: A Comprehensive Review"

_processes, doi:10.3390/pr11030731_

Round 1

Reviewer 1 Report

The abstract is not in line with the title. It should summarize the objectives of the article, methodology, results and conclusions.

There is no reference to Pakistan in the title or abstract. Yet, the introduction starts with Pakistan. If the authors want to address an international audience, they should present arguments available for the entire research community.

The objectives defined in the introduction are not in line with the methodology of the paper. Also, the way that the objectives are met by this work is not presented. There is only one reference to the objectives in this paper.

The presentation form is not adequate. There are large tables and minor explanations for them. There is no theoretical framework supporting the work.

The review does not have the quality of a research paper, in my opinion, but rather of a book chapter or lecture. There is no contribution to the knowledge.

There should be a discussion of the results.

Author Response

All authors are thankful for your positive review and comments; After analyzing your comments and the manuscript, all the said changes are made with the permission of all co-authors. 

Reviewer 2 Report

Manuscript Processes – 2013970

The manuscript “Composting Processes for Solid Waste Management: A Comprehensive Review” brings an overview of the current composting processes for solid waste manage providing valuable information about the mechanism, the factors that affect the process, and future perspectives. Authors also compare conventional composting processes with the innovative two-stage method composing. There are a few comments to be answered before acceptance:

General comments:

1.     The aim b) of the review does not fit with the content of the manuscript. Although the authors discuss composting methods in section 6. Conclusion and recommendations, authors should provide a more critical and comparative analysis (conclusion) with the advantages and drawbacks that helps to select the most appropriate treatment, according to objective b).

2.     Figure 1 does not provide relevant information; Authors can remove it, to reduce the total number of Figures, which is excessive.

3.     From Table 1 it is difficult to draw a conclusion; it is difficult to read. It does not state what is written in the text on line 134. If the purpose is to collect a comparison of methods, the Table should highlight the differences and advantages between them. Authors can make Table 1 clearer, for example, highlighting the methods and summarizing the key parameters.

4.     In section 2.2. Discrete waste's composting mentions that persistent organic pollutants and endocrine disruptors are reduced but does not explain how. It would be interesting to describe how and why those contaminants are reduced in composting, the processes that are involved, at least briefly.

5.     Section 4 is addressed only to the two-stage techniques, specifically AnC followed by AC. Therefore, the section should be reestructured, and section 4.1 should be eliminated.

6.     Line 544 - 553. This has not been commented on previously in section 4.1. Authors should comment on it before, instead of in conclusions.

Specific comments:

Line 56: “These actions causing rodent growth, emitting offensive odors, and hideous insects, leading” Rephrase, what are the rodents and insects that cause global warming?

Line 58: For instance

Line 63: Are used

Line 64: “…the composting intends to avoid ground water from being polluted because composting reduces…  Redundant. Rephrase

Line 88: Figure 2 without parentheses

Line 96: A few composting processes and frameworks have been created, changing from miniature home-made reactors utilized by individual families over medium-sized on-site reactors worked by ranchers to expansive, high-tech reactors used by competent compost makers. Rephrase, it is not understandable.

Line 117, 121, 132: Mentioning figure 3 many times is not necessary.

Table 3: lowercase pH. The value is not a range, it is not correct to express it like this.

Line 455: Subdivide section 2.2. in section 2.2.1., 2.2.2…

Line 515. “…were anaerobic and anaerobic digestion,…” Both were anaerobic?

Line 531: Figure 8 without parentheses

English language needs to be reviewed.

Author Response

(The authors gave the same response as above.)

Reviewer 3 Report

This is an interesting Review, that collects and presents composting processes for solid waste management. The interested reader finds information about  the different aspects and problems of composting.
The topic is clearly discussed. However, Tables may be imroved. For example, Table 1 is very difficult to read. Probably it is better to insert it in an horizontal rather than vertical layout
Attention! In Figure 2 (center, above) it is written "Green (Grass, food scarps, manures)"
According to English Dictionary, scarp = a steep slope or cliff formed by folded or eroded layers of rock. Of course, this is not the case here.

Maybe it was misspelled instead of "scraps"

Author Response

(The authors gave the same response as above.)

Round 2

Reviewer 1 Report

Rows 54-55: There is a repetition. The sentence makes no sense.

Rows 63-67: This is definitely not an academic level. “Insects are hideous”?

There are more mistakes than in the first form of the article. Many phrases make no sense (see, for instance, rows 116 – 119, rows 169 – 170).

In my opinion, the paper does not bring any contribution. Also, the changes brought to the initial manuscript are minor and don’t bring additional value to the text.

Author Response

(The authors gave the same response as above.)

Round 3

Reviewer 1 Report

As mentioned earlier, the study needs to be significantly improved in order to be considered. The version that you sent right now does not include significant changes. All the changes are made in the introduction and conclusions.

Before making any further changes to this manuscript, please check the Instructions for authors (https://www.mdpi.com/journal/processes/instructions). The requirements for the reviews are the following, for this particular journal:

  • Review manuscripts should comprise the front matter, literature review sections and the back matter. The template file can also be used to prepare the front and back matter of your review manuscript. It is not necessary to follow the remaining structure. Structured reviews and meta-analyses should use the same structure as research articles and ensure they conform to the PRISMA guidelines.

Everything you need to know about PRISMA is here: https://prisma-statement.org/PRISMAStatement/

I really hope that you will find the 

  • PRISMA 2020 checklist
  • PRISMA 2020 flow diagram
  • etc. extremely useful!

Unfortunately, right now this work is not conducted in accordance with the journal's requirements. 

Author Response

All authors are thankful for your positive review and comments; After analyzing your comments and the manuscript, all the said changes are made with the permission of all co-authors. We have amended all data which you mentioned in the review and recreated the same data according to your instruction in the previous comment. We changed some sections completely, some changed parts, and a few paragraphs have slight changes. Review the references section and create an accurate bibliography according to your guidelines and remove the irrelevant references as well.
